# FTSO: Effective NAS via First Topology Second Operator

## Abstract

Existing one-shot neural architecture search (NAS) methods generally contain a giant super-net, which leads to heavy computational cost. Our method, named FTSO, separates the whole architecture search into two sub-steps. In the first step, we only search for the topology, and in the second step, we only search for the operators. FTSO not only reduces NAS's search time from days to 0.68 seconds, but also significantly improves the accuracy. Specifically, our experiments on ImageNet show that within merely 18 seconds, FTSO can achieve 76.4% testing accuracy, 1.5% higher than the baseline, PC-DARTS. In addition, FTSO can reach 97.77% testing accuracy, 0.27% higher than the baseline, with 99.8% of search time saved on CIFAR10.

## 1 Introduction

Since the great success of AlexNet (Krizhevsky et al., 2012) in image classification, most modern machine learning models have been developed based on deep neural networks. For neural networks, their performance is greatly determined by the architectures. Thus, in the past decade, a tremendous amount of work (Simonyan & Zisserman, 2015; Szegedy et al., 2015; He et al., 2016) has been done to investigate proper network architecture design. However, as the network size has grown larger and larger, it has gradually become unaffordable to manually search for better network architectures via trial and error due to the expensive time and resource overhead. To ease this problem, a new technique called neural architecture search (NAS) was introduced, which allows computers to search for better network architectures automatically instead of relying on human experts..

Early-proposed reinforcement learning-based NAS methods (Zoph & Le, 2017; Baker et al., 2017; Zoph et al., 2018) typically have an RNN-based controller to sample candidate network architectures from the search space. Although these algorithms can provide promising accuracy, their computation cost is usually unaffordable, e.g., 1800 GPU-days are required for NASNet.

To ease the search efficiency problem, one-shot approaches (Pham et al., 2018; Cai et al., 2019; Liu et al., 2019) with parameter sharing were proposed. These methods first create a huge directed acyclic graph (DAG) super-net, containing the whole search space. Then, the kernel weights are shared among all the sampled architectures via the super-net. This strategy makes it possible to measure the candidate architecture's performance without repeatedly retraining it from scratch. However, these algorithms suffer from the super-nets' computational overheads. This problem is particularly severe for differentiable models (Liu et al., 2019; Xu et al., 2020).

Limited by current NAS algorithms' inefficiency, it is rather challenging to find satisfying network architectures on large-scale datasets and high-level tasks. For instance, current speed-oriented NAS approaches generally require days to accomplish one search trial on ImageNet, e.g., 8.3 GPU-days for ProxylessNAS (Cai et al., 2019) and 3.8 GPU-days for PC-DARTS (Xu et al., 2020). Therefore,we argue that it is essential to propose a new well-defined search space, which is not only expressive enough to cover the most powerful architectures, but also compact enough to filter out the poor architectures.

Motivated by Shu et al. (2020), who demonstrated that randomly replacing operators in a found architecture does not harm the accuracy much, we believe that it could not only bring no reduction to the testing accuracy but also significantly benefit the search efficiency if we omit the influence of operators and cluster architectures according to the topology. Thus, in this paper, we propose to

separately search for the network topology and the operators. We name this new method Effective NAS via First Topology Second Operator (FTSO).

In this paper, we mathematically prove that FTSO reduces the required parameters by $5.3 \times 10^7$ and decreases the FLOPs per iteration by $1 \times 10^5$. Besides, FTSO significantly promotes the accuracy compared to the baseline by greatly shrinking the search space, reducing the operators complexity in magnitude and lowering the required searching period from 50 epochs to one iteration. What's more, the Matthew effect is eased.

Furthermore, we empirically show that FTSO is superior in both efficiency and effectiveness, accomplishing the whole architecture search in $0.68$ seconds. On ImageNet, FTSO can achieve $76.4\%$ testing accuracy, $1.5\%$ higher than the baseline, within merely 18 seconds. More importantly, when we only search for one iteration, FTSO can reach $75.64\%$ testing accuracy, $0.74\%$ higher than the baseline in just $0.68$ seconds. Besides, if we allow FTSO to search for 19 minutes, $76.42\%$ Top1 and $93.2\%$ Top5 testing accuracy can be obtained. In addition, FTSO can reach $97.77\%$ testing accuracy, $0.27\%$ higher than the baseline, with $99.8\%$ of search time saved on CIFAR10. Although in this paper we have only implemented FTSO within a continuous search space, we illustrate in Section 5 that FTSO can be seamlessly transferred to other NAS algorithms.

## 2    RELATED WORK

In general, existing NAS algorithms can be divided into three categories, namely, reinforcement learning-based, revolution-based and differentiable. Early-proposed reinforcement learning-based methods (Zoph & Le, 2017; Zoph et al., 2018) generally suffer from high computational cost and low-efficiency sampling. Instead of sampling a discrete architecture and then evaluating it, DARTS (Liu et al., 2019) treats the whole search space as a continuous super-net. It assigns every operator a real number weight and treats every node as the linear combination of all its transformed predecessors. To be specific, DARTS's search space is a directed acyclic graph (DAG) containing two input nodes inherited from previous cells, four intermediate nodes and one output node. Each node denotes one latent representation and each edge denotes an operator. Every intermediate node $\mathbf{x}_j$ is calculated from all its predecessors $\mathbf{x}_i$, i.e., $\mathbf{x}_j = \sum_{i<j} \sum_{o \in \mathcal{O}} \frac{\exp \alpha_{i,j}^o}{\sum_{o' \in \mathcal{O}} \exp \alpha_{i,j}^{o'}} o(\mathbf{x}_i)$, where $\mathcal{O}$ denotes the collection of all candidate operators, $\alpha_{i,j}^o$ denotes the weight for operator $o$ from node $i$ to $j$. This strategy allows DARTS to directly use gradients to optimize the whole super-net. After the super-net converges, DARTS only retains the operators with the largest weights. In this way, the final discrete architecture is derived. The main defect of DARTS is that it needs to maintain and do all calculations on a giant super-net, which inevitably leads to heavy computational overheads and over-fitting.

Proposed to relieve the computational overhead of DARTS, DARTS-ES (Zela et al., 2020) reduces the number of searching epochs via early stopping, according to the Hessian matrix's max eigenvalue. PC-DARTS (Xu et al., 2020) decreases the FLOPs per iteration by only calculating a proportion of the input channels and retaining the remainder unchanged, and normalizes the edge weights to stabilize the search. To be specific, in PC-DARTS, every intermediate node $\mathbf{x}_j$ is computed from all its predecessors $\mathbf{x}_i$, i.e., $\mathbf{x}_j = \sum_{i<j} \frac{\exp \beta_{i,j}}{\sum_{i'<j} \exp \beta_{i',j}} f_{i,j}(\mathbf{x}_i)$, where $\beta_{i,j}$ describes the input node $i$'s importance to the node $j$, and $f_{i,j}$ is the weighted sum of all the candidate operators' outputs between node $i$ and $j$. Specifically, $f_{i,j}(\mathbf{x}_i, \mathbf{S}_{i,j}) = \sum_{o \in \mathcal{O}} \frac{e^{\alpha_{i,j}^o}}{\sum_{o' \in \mathcal{O}} e^{\alpha_{i,j}^{o'}}} o(\mathbf{S}_{i,j} * \mathbf{x}_i) + (1 - \mathbf{S}_{i,j}) * \mathbf{x}_i$, where $\mathbf{S}_{i,j}$ denotes a binary vector, in which only $1/K$ elements are 1.

## 3    FTSO: EFFECTIVE NAS VIA FIRST TOPOLOGY SECOND OPERATOR

Existing NAS approaches generally suffer from the heavy computational overhead and the unsatisfying testing accuracy leaded by the huge search space. Such problems are especially stern in one-shot and differentiable methods because, these algorithms need to maintain and even do all the calculations directly on the search space.

To ease such problems, it is of great demand to investigate the correlations among different architectures and to shrink the search space according to these prior knowledge. We notice that there

is an important observation in Shu et al. (2020) that randomly substituting the operators does not observably influence the testing accuracy in a found architecture. Therefore, it would be a great inspiration that we could cluster the architectures according to their connection topologies. To be specific, suppose we have found an architecture only containing the simplest operators achieves high accuracy on the testing set, if we replace all the skip connections in this architecture with powerful operators, the converted architecture can also perform well on the testing set with high confidence.

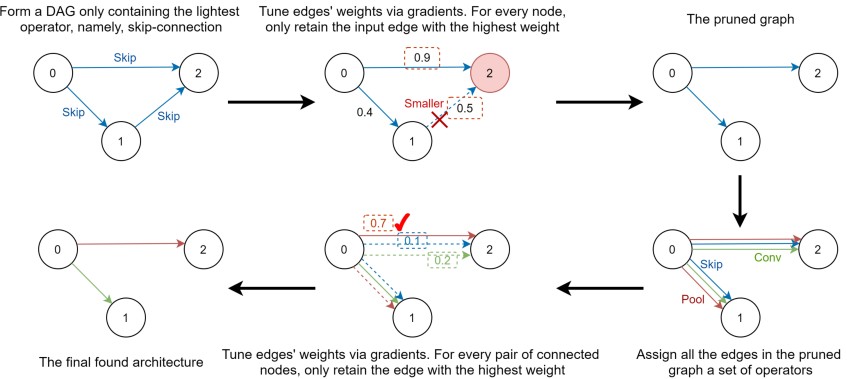

Figure 1: The main structure of FTSO.

Thus, in this paper, we first propose to find the effective network topology with simple operators and then we fix the topology, and search for the most suitable operators for the given topology. In this way, the testing accuracy can still be guaranteed, while the search space is shrunk in magnitude. We name this new NAS algorithm Effective NAS via First Topology Second Operator (FTSO).

As shown in Figure 1, we inherit the differentiable framework of PC-DARTS, and divide the architecture search into two phases. We name these two phases topology search and operator search, and illustrate how they work in Algorithm 1 and 2, respectively. In the first phase, we form a super-net only containing the simplest operator, *skip connection*. Because the *skip connection* operator contains no kernel weights, we only need to optimize the architecture parameters $\beta_{i,j}$, denoting the node $i$'s importance to the node $j$. In fact, as shown in Table 3, the *max pooling* operator also brings satisfying results for the topology search. There are two reasons why we use *skip connection*. The first reason is that the *skip connection* operator not only requires zero parameter, but also produces the minimum computational cost. The second reason is that max pooling may lead to the loss of useful information if the network is deep. Furthermore, as the the only difference between our topology search and the vanilla DARTS is the number of candidate operators, the pruned architecture's connectivity can be guaranteed.

After the topology search, for every intermediate node $j$, we only retain its connections to its predecessors $i^*$ with the highest two $\beta_{i,j}$ similar to DARTS and PC-DARTS. In the second phase, we search for the operators suitable for the pruned topology with two strategies. One replaces each operator in the pruned topology with a mix-operator $f_{i,j}$, where $f_{i,j}$ is the linear combination of all the candidate operators $o$ with weight $\alpha_{i,j}^o$, denoting the importance of the operator $o$ between the node $i$ and $j$. After that, we optimize the architecture parameters, $\alpha_{i,j}^o$, $\beta_{i,j}$ and the kernel weights $\omega_{i,j}^o$ alternatively. After the super-net converges, we only retain one operator $o^*$ with the highest $\alpha_{i,j}^o$ for every two connected nodes $i$ and $j$. The other is to directly replace all the operators in the pruned topology with one single operator owning the highest model capacity, e.g., a convolution operator. In this paper, we take the second strategy as the default configuration because it is much more efficient and avoid over-fitting. The final architecture outputted by the first strategy is only a small part of the super-net. If the sub-graph can generalize perfectly on the testing set, the super-net must over-fit. Thus, under this circumstance, the optimum super-net on training set is not the optimum one on testing set, and so is the sub-graph. PC-DARTS has mentioned this issue and reduced the size of the super-net via only computing partial input channels. While in the second strategy, since no super-net is adopted and all the simple operators in the sub-graph are replaced with powerful

operators, the final architecture's model capacity gets promoted. Additional empirical comparisons between these two strategies can be found in Section 4.4.

---

**Algorithm 1:** topology search

**Input:** a set of nodes: $n_k$
**Output:** the pruned architecture: $A_p$
1 Create an directed edge $e_{i,j}$ with weight $\beta_{i,j}$ between each pair of nodes $n_i$ and $n_j$ ($i < j$);
2 Assign each edge $e_{i,j}$ a *skip connection* operator $o_{i,j}$ with kernel weights $w_{i,j}$;
3 **while** *still in the first epoch* **do**
4     Forward-propagate following $n_j = \sum_{i<j} o(n_i)\beta_{i,j}$;
5     Update architecture $\beta$ by descending $\nabla_\beta \mathcal{L}_{val}(w, \beta)$;
6     Update weights $w$ by descending $\nabla_w \mathcal{L}_{train}(w, \beta)$;
7 **foreach** *node* $n_j \in A_p$ **do**
8     $T_j \leftarrow$ the second largest $\beta_{i,j}$;
9     **foreach** *node* $n_i$ **do**
10        **if** $\beta_{i,j} < T_j$ **then**
11           Prune edge $e_{i,j}$
12 Derive the pruned architecture $A_p$.

---

**Algorithm 2:** operator search

**Input:** the pruned architecture produced by the topology search: $A_p$
**Output:** the found architecture: $A_f$
1 **if** *replace with convolutions* **then**
2     Replace all the retained operators $o_{i,j}$ in $A_p$ with convolutions;
3 **else**
4     Assign each node $n_j \leftarrow \sum_{i<j} \sum_{o \in \mathcal{O}} \frac{\exp \alpha_{i,j}^o}{\sum_{o' \in \mathcal{O}} \exp \alpha_{i,j}^{o'}} o(n_i)$;
5     **while** *not converged* **do**
6        Update architecture $\alpha$ by descending $\nabla_\alpha \mathcal{L}_{val}(w, \alpha)$;
7        Update weights $w$ by descending $\nabla_w \mathcal{L}_{train}(w, \alpha)$;
8 **foreach** *edge* $e_{i,j} \in A_p$ **do**
9     Assign edge $e_{i,j}$ the operator $o' \in \mathcal{O}$ with the highest $\alpha_{i,j}^{o'}$;
10 Derive the found architecture $A_f \leftarrow A_p$.

---

In DARTS, the network topology and operators are jointly searched, which makes both the size and the computational cost of the super-net extremely high. We use $n$ to denote the number of nodes, $p$ to denote the number of candidate operators. Since we have two input nodes, one output node and $n - 3$ intermediate nodes, the super-net contains totally $\frac{1}{2}(n^2 - 3n)$ edges. At the same time, every edge keeps $p$ operators, thus, the total number of operators in DARTS is $\frac{1}{2}(n^2 - 3n)p$. By comparison, there are only $\frac{1}{2}n(n-3)$ operations in our topology search, and $2(n-3)p$ operations in our operator search. This is because in the topology search, every edge contains only one operator; and in the topology search, every intermediate node only connects to two predecessors. Since $n$ is usually close to $p$, FTSO reduces the number of operations from $O(n^3)$ to $O(n^2)$.

In addition to the reduction on the number of operations, FTSO also dramatically decreases the internal cost of the operations because, during the topology search, all the powerful operators are replaced with the simple operators. To be specific, a vanilla convolutional operator needs $k^2 C_{in} H_{out} W_{out} C_{out}$ FLOPs and $(k^2 C_{in} + 1)C_{out}$ parameters, where $k$ is the kernel size, $C_{in}$ is the input tensor's channel number and $H_{out}, W_{out}$, and $C_{out}$ are the output tensor's height, width and channel number respectively. By comparison, a *skip connection* operator needs 0 parameters and 0 FLOPs. For simplicity, assume all the candidate operators are convolutions. Since DARTS has $\frac{1}{2}pn(n-3)$ edges, it needs to compute $\frac{1}{2}pn(n-3)$ convolutions and $\frac{1}{2}pn(n-3)$ tensor summations. Because each tensor summation consumes $H_{in} W_{in} C_{in}$ FLOPs, DARTS requires totally $\frac{1}{2}pk^2 n(n-3)C_{in} H_{out} W_{out} C_{out} + \frac{1}{2}pn(n-3)H_{out} W_{out} C_{out} = \frac{1}{2}pn(n-3)H_{out} W_{out} C_{out}(k^2 C_{in} + 1)$ FLOPs

and $\frac{1}{2}n(n-3)p(k^2C_{in}+1)C_{out}$ parameters. While in FTSO, if we first search for the topology and then directly substituting the operators, only $\frac{1}{2}n(n-3)$ tensor summation need to be calculated. Thus, the total number of FLOPs and parameters of FTSO are $\frac{1}{2}n(n-3)H_{in}W_{in}C_{in}$ and $\frac{1}{2}n(n-3)$ respectively. As a typical configuration, let $k = 5$, $C_{in} = C_{out} = 512$, $n = 7$, $p = 8$. Then, our algorithm requires only $\frac{1}{p(k^2C_{in}+1)C_{out}} = 1.9 \times 10^{-8}$ times the parameters and $\frac{1}{p(k^2C_{in}+1)} = 9.8 \times 10^{-6}$ times the forward-propagation FLOPs per iteration compared to those of DARTS.

FTSO's huge reduction on the parameter numbers provides us a large amount of benefits. As mentioned above, it allows the algorithm to converge in only a few iterations and prevent over-fitting. This is because when extracting the discrete sub-graph from the super-net, many architecture parameters are set to be 0. The introduced disturbance impacts more on the over-fitting super-nets since they prefer sharper local minimums. Furthermore, it avoids the Matthew's effect. Each architecture has only one iteration to tune its kernel weights in DARTS. However, within one iteration, only the operators with few parameters can converge and thus, the simpler operators outperform the powerful ones in the super-net, then obtain larger gradients to enlarge their advantages. In this way, the found architectures tend to only contain the simplest operators and perform poorly on both training and testing set. Since FTSO only contains one operator with 0 parameter, the Matthew's effect is eliminated.

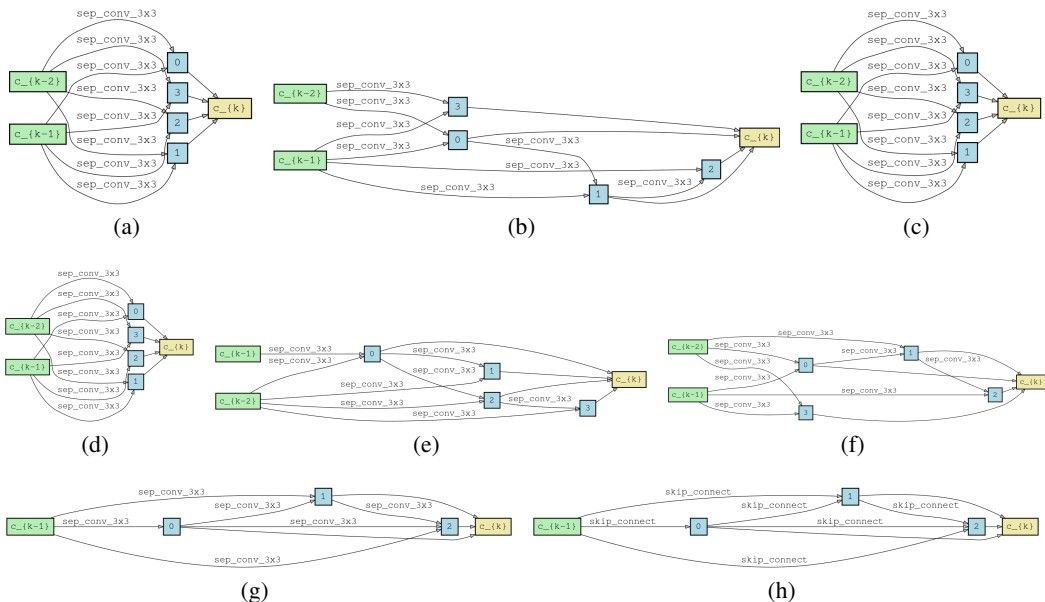

Figure 2: FTSO's found architectures. (a) and (b): Normal and reduction cells found on CIFAR10 after one epoch's search; (c) and (d): Normal and reduction cells found on the entire ImageNet after one epoch's search; (e) and (f): Normal and reduction cells found on CIFAR10, where we adopt the operator search, and use the $3 \times 3$ *separable convolution* to search for the topology; (g): FTSO's cell found on NATS-Bench; (h): DARTS's cell found on NATS-Bench.

## 4 EXPERIMENTS

Our search algorithm is evaluated on the three most widely-used datasets in NAS papers, namely, CIFAR10(Krizhevsky et al., 2009) and ImageNet(Russakovsky et al., 2015) and NATS-Bench(Dong et al., 2020). Following DARTS, our search space contains a total eight candidate operators: $3 \times 3$ and $5 \times 5$ *separable convolutions*, $3 \times 3$ and $5 \times 5$ *dilated separable convolutions*, $3 \times 3$ *max* and *average pooling*, *skip connection* (i.e., $output = input$) and *zero* (i.e., $output = 0$). When searching for the topology, we pick only one operator from the candidate set. As mentioned in Section 3, we have two strategies to determine the operators, including the one based on gradients and the one is directly replacing. In Section 4.1 and 4.2, we focus more on the second configuration. In Section

4.4, we have a comprehensive comparison on these two strategies. All detailed configurations are shown in Table 4 in Appendix A.1. In addition, most of our experiments only search for one epoch or one iteration because of the benefits of FTSO's huge reduction on the parameter numbers. For more experimental supports, please refer to Section 4.4. Note that it is almost impossible for the existing models to obtain satisfying results via only searching for one epoch or one iteration because their super-nets contain a large amount of parameters, which require a long period to tune.

## 4.1 RESULTS ON CIFAR10

We compare FTSO to existing state-of-the-art NAS methods in Table 1. In the experiment, we only search for the topology with skip-connections and then replace them all with $3 \times 3$ *separable convolutions*. The reason that we do not adopt $5 \times 5$ *separable convolutions* is that the pre-processed input images do not have enough resolutions, and the network is rather deep. After a few layers, the convolution's receptive field has been larger than the whole image. At that time, larger convolutional kernels may not bring benefits. Instead, the extra parameters brought by the larger kernel size may lead to over-fitting. On the other hand, suppose both the image's resolution and the dataset's scale are big enough and the evaluation period is adequate, the $5 \times 5$ *separable convolution* might be a better choice. The found architecture after one epoch's search is shown in Figure 2. Because FTSO contains only a few trainable parameters, it can even achieve comparable accuracy to PC-DARTS with only an one-time gradient update. Under this configuration, merely $0.68$ seconds are required and $99.993\%$ of the search time is saved. In addition, as shown in Table 3, when the topology is searched with powerful operators for a long period, additional operator search usually helps. However, when we search for the topology with simple operators for a short period, omitting the operator search may lead to better results. This is because with simple operators and very few updates, the found topology can already generalize quite well.

Table 1: Comparison with existing state-of-the-art image classification architectures

| Architecture | CIFAR Err.(%) | | ImageNet Err.(%) | | Search Cost (GPU-days) | |
|---|---|---|---|---|---|---|
| | 600 | 1200 | Top1 | Top5 | CIFAR | I.N. |
| NASNet-A[†](Zoph et al., 2018) | 2.65 | - | 26.0 | 8.4 | 1800 | - |
| AmoebaNet[†](Real et al., 2019) | 2.55 | - | 24.3 | 7.6 | 3150 | - |
| PNAS(Liu et al., 2018) | 3.41 | - | 25.8 | 8.1 | 225 | - |
| ENAS[†](Pham et al., 2018) | 2.89 | - | - | - | 0.5 | - |
| DARTS (2nd)[†](Liu et al., 2019) | 2.76 | - | 26.7 | 8.7 | 1 | 4.0 |
| SNAS[†](Xie et al., 2019) | 2.85 | - | 27.3 | 9.2 | 1.5 | |
| ProxylessNAS[†*](Cai et al., 2019) | 2.08 | - | 24.9 | 7.5 | 4.0 | 8.3 |
| P-DARTS[†](Chen et al., 2019) | 2.50 | - | 24.4 | 7.4 | 0.3 | - |
| BayesNAS[†](Zhou et al., 2019) | 2.81 | - | 26.5 | 8.9 | 0.2 | - |
| PC-DARTS(CIF.)[†](Xu et al., 2020) | 2.57 | 2.50 | 25.1 | 7.8 | 0.1 | - |
| PC-DARTS(I.N.)[†*](Xu et al., 2020) | - | - | 24.2 | 7.3 | - | 3.8 |
| FTSO (CIFAR10 + 1 epoch)[†] | 2.48 | **2.23** | 23.60 | 7.01 | $2 \times 10^{-4}$ | - |
| FTSO (CIFAR10 + 1 iteration)[†] | 2.68 | 2.54 | 24.36 | 7.27 | $\mathbf{7.87 \times 10^{-6}}$ | - |
| FTSO (Full ImageNet + 1epoch)[†*] | **2.35** | 2.26 | **23.58** | **6.80** | - | **0.01** |

[†] When testing on CIFAR10, these models adopt cut-out.
[*] These models are directly searched on ImageNet.

## 4.2 RESULTS ON IMAGENET

On ImageNet, we use similar configurations to those on CIFAR10. When searching, we have two configurations. The detailed configurations are shown in Appendix A.1. Our experiments in Table 1 show that FTSO is significantly superior to existing methods in both efficiency and effectiveness. The found architectures after one epoch's search on CIFAR10 and the entire ImageNet are shown

in Figure 2 . It is surprising that the best architecture we found on ImageNet is the shallowest and widest one. Compared to the much more "reasonable" architectures shown in Figure 2(e) and 2(f), which was found with the topology search only containing $3 \times 3$ separable convolutions and an additional operator search on CIFAR10, the "abnormal" architecture, containing the same amount of FLOPs and parameters, can achieve $0.78\%$ higher testing accuracy. We think this is because the whole model is stacked with many cells. If the depth of each cell is too high, it leads to a very deep neural network. At that time, because all the operators in our found architecture are convolutions, we cannot use skip connections to facilitate gradients' propagation in ResNet's manner. In this way, both the vanishing and explosion of gradients may prevent the deeper models from higher performance.

### 4.3 RESULTS ON NATS-BENCH

In the search space of NATS-Bench, there are one input node, three intermediate nodes and one output node, and each intermediate node connects to all its predecessors. Here we implement FTSO based on DARTS instead of PC-DARTS, and we compare FTSO's performance to other NAS algorithms in Table 2. It is shown that FTSO dominates DARTS on all configurations. Coinciding with our analysis in Section 3, the architectures found by DARTS tend to only contain simple operators, thus cannot achieve satisfying accuracy. For example, when searching on CIFAR10, the architecture found by DARTS is full of *skip connections* as shown in Figure 2(h). By comparison, as shown in Figure 2(g), the architecture found by FTSO is much more powerful.

Table 2: Comparison with existing state-of-the-art image classification architectures

| Architecture | Search on CIFAR10 | | | Search on CIFAR100 | | | Search on ImageNet | | |
|---|---|---|---|---|---|---|---|---|---|
| | CF10 | CF100 | I.N. | CF10 | CF100 | I.N. | CF10 | CF100 | I.N. |
| DARTS (1st) | 54.30 | 15.61 | $16.32^{\dagger}$ | 86.57 | 58.13 | 28.50 | 89.58 | 63.89 | 33.77 |
| DARTS (2nd) | 86.88 | 58.61 | 28.91 | 91.96 | 67.27 | 39.47 | 84.64 | 55.15 | 26.06 |
| FTSO | 93.98 | 70.22 | 45.57 | 93.98 | 70.22 | 45.57 | 93.98 | 70.22 | 45.57 |

$^{\dagger}$ This means that within NATS-Bench's search space, when we use the 1st order DARTS to search for architectures on CIFAR10 dataset, the found architecture can achieve $16.32\%$ testing accuracy on ImageNet.
$^{*}$ CF10 means testing accuracy (%) on CIFAR10; CF100 means testing accuracy (%) on CIFAR100; I.N. means testing accuracy (%) on ImageNet.

### 4.4 ABLATION STUDY

In terms of topology-only search, one epoch is just enough, thanks to the many fewer kernel weights contained in FTSO, and more search epochs bring obvious disadvantages because of over-fitting. Since one epoch performs better than more epochs, it raises the question whether one iteration is also superior to more iterations. In Figure 3(c) we show that although one iteration cannot surpass one epoch, it is better than a few iterations. This is because when we only search for one iteration, the model does over-fit the data, thus the model generalizes well. When we only searched for a few iterations, the number of different images seen by the model is not big enough. However, since the super-net only contains skip connections, such number of gradient updates has been enough for the architecture parameters to get over-fitted. This is the reason that a few iterations perform worse than one iteration. After we have searched for one whole epoch, the super-net has seen enormous different images, this helps it to generalize better on the testing set. This is the reason that one epoch performs the best. In terms of whether we should search for one iteration or two, in Figure 3(d), we show that both choices work well.

When we do not search for the operators after the topology search, we assign all the remaining edges a fixed operator. Thus, which operator we should choose becomes a critical question. Figure 3(e) show that $3 \times 3$ *separable convolution* can indeed outperform all other operators in terms of accuracy.

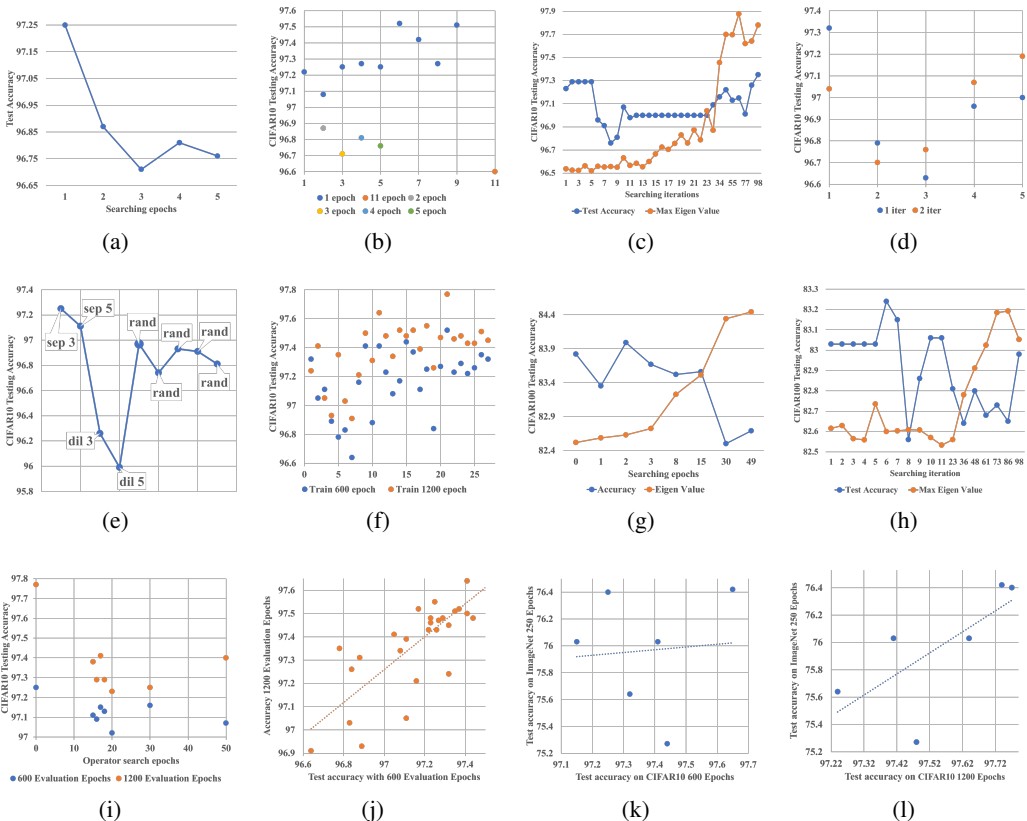

Figure 3: Ablation study. (a): CIFAR10: Accuracy - Search epochs (in the same run); (b): CIFAR10: Accuracy - Search epochs (multiple runs); (c): CIFAR10: Accuracy - Search iterations; (d): Accuracy on CIFAR10: 1 vs 2 search iterations (multiple runs); (e): CIFAR10: Accuracy - Operator replacing skip-connections; (f): CIFAR10: Accuracy - Training epochs of the found architecture; (g): Epoch-wise on CIFAR100: Accuracy - Max eigenvalue $\nabla^2_{Arch}\mathcal{L}_{val}$; (h): Iteration-wise on CIFAR100: Accuracy - Max eigenvalue $\nabla^2_{Arch}\mathcal{L}_{val}$; (i): CIFAR10: Operator search's epoch-wise impact on testing accuracy (0 epoch means only searching for the topology); (j): CIFAR10: The found architecture's training epochs' impacts on the testing accuracy (same architecture); (k):Testing accuracy: CIFAR10 600 evaluation epochs - ImageNet 250 evaluation epochs (evaluating after training the same architecture on different datasets); (l): Testing accuracy: CIFAR10 1200 evaluation epochs - ImageNet 250 evaluation epochs (evaluating after training the same architecture on different datasets)

As shown in Figure 3(f), we find that under a different number of evaluation epochs, both the absolute value and the relative ranking of the same architecture's testing accuracy may vary; i.e., some architectures performing well within 600 epochs perform poorly after 1200 epochs. However, in general, they still obey a positive correlation, with a Pearson correlation coefficient of 0.77, as shown in Figure 3(j). In terms of the generalization ability from CIFAR10 to ImageNet, Figure 3(l) reveals that the architectures performing well after long-term evaluation on CIFAR10 can usually generalize better on ImageNet, with the correlation coefficient of 0.7; yet, as shown in Figure 3(k), there is no guarantee that those working well on CIFAR10 within limited evaluation epochs can also dominate on ImageNet because this only proves that they can quickly converge, not that they can converge to a global optimum.

In Figure 3(g) and 3(h), we show that one epoch is not only an optimal choice on CIFAR10, but also enough for the topology-only search on CIFAR100. In addition, as the search epochs and iterations increase, the max eigenvalue of the loss's Hessian matrix on the validation set increases and the testing accuracy is generally decreasing because the model's generalization ability is dropping. This phenomenon is particularly obvious epoch-wise because, after just a few iterations, the model can

Table 3: Different configurations' impacts to FTSO

| FTSO's Configuration | CIFAR10 Err.(%) | | ImageNet Err.(%) | | Search Cost (GPU-days) | |
|---|---|---|---|---|---|---|
| | 600 | 1200 | Top1 | Top5 | CIFAR | ImageNet |
| CIF. Topo(skip,1it.) | 2.68 | 2.54 | 24.36 | 7.27 | $7.87 \times 10^{-6}$ | - |
| CIF. Topo(skip,1ep.) | 2.48 | 2.23 | 23.60 | 7.01 | $2 \times 10^{-4}$ | - |
| CIF. Topo(skip,50ep.) | 2.77 | 2.52 | - | - | 0.01 | - |
| CIF. Topo(skip,1ep.)+Op(18ep.) | 2.85 | 2.59 | 23.97 | 7.20 | 0.01 | - |
| CIF. Topo(skip,50ep.)+Op(50ep.) | 2.59 | 2.36 | 23.97 | 7.12 | 0.05 | - |
| CIF. Topo(m.p.,50ep.)+Op(50ep.) | 2.83 | 2.48 | - | - | 0.05 | - |
| CIF. Topo(sep3,50ep.) | 2.63 | 2.48 | - | - | 0.02 | - |
| CIF. Topo(sep3,50ep.)+Op(50ep.) | 2.56 | 2.52 | 24.73 | 7.60 | 0.06 | - |
| CIF. Topo(3op,50ep.)+Op(50ep.) | 2.59 | 2.50 | - | - | 0.05 | - |
| CIF. Topo(4op,50ep.)+Op(50ep.) | 2.68 | 2.59 | - | - | 0.07 | - |
| Part ImageNet Topo(skip,1it.) | - | - | 24.03 | 7.07 | - | 0.0002 |
| Part ImageNet Topo(skip,1ep.) | - | - | 23.94 | 7.05 | - | 0.0017 |
| Part ImageNet Topo(skip,6ep.) | - | - | 24.59 | 7.38 | - | 0.009 |
| Full ImageNet Topo(skip,1ep.) | 2.35 | 2.26 | 23.58 | 6.80 | - | 0.01 |

[*] '3op' means *max pool 3x3*, *skip connect* and *none*; '4op' means: *sep conv 3x3*, *max pool 3x3*, *skip connect* and *none*; 'm.p.' means: *max pool 3x3*; 'sep3' means: *sep conv 3x3*.

[*] 'CIF.' means CIFAR10; 'Topo(skip,1it)' means to search for topology with only skip connections for 1 iteration; '1ep' means 1 epoch; 'Part ImageNet' means to search on part of ImageNet.

already reach a comparable accuracy on the training set. Then the model's performance on the testing set starts to relate with its generalization ability.

## 5 GENERALIZATION TO OTHER TASKS AND SEARCH SPACES

As shown in Section 4, FTSO works well under different search spaces and node numbers. Theoretically, FTSO's advantages to DARTS can be enlarged while search space and node number increase. This is because FTSO reduces the computational cost from $O(n^3)$ to $O(n^2)$ and avoids over-fitting. Although in this paper, we establish FTSO within differentiable search spaces, in fact, the first topology second operator strategy is not limited within any specific search space or tasks. No matter the search space is discrete or continuous, we first shrink the candidate operator set, and only retain the simplest operator. After this, the size of the whole search space is reduced in magnitude. Then, we search for the best topology with whatever search algorithm. In this way, a promising topology can be found. Then, we can either directly assign each edge a powerful operator or use gradients to search for operators. Generally, the directly replacing strategy leads to higher accuracy, and the gradient-based strategy reduces model complexity.

## 6 CONCLUSION

In this paper, we propose an ultra computationally efficient neural architecture search method named FTSO, which reduces NAS's search time cost from days to less than 0.68 seconds, while achieving 1.5% and 0.27% accuracy improvement on ImageNet and CIFAR10 respectively. Our key idea is to divide the search procedure into two sub-phases. In the first phase, we only search for the network's topology with simple operators. Then, in the second phase, we fix the topology and only consider which operators we should choose.

Our strategy is concise in both theory and implementation, and our promising experimental results show that current NAS methods contain too much redundancy, which heavily impacts the efficiency and becomes a barrier to higher accuracy. What is more, our method is not bound by differentiable search spaces, it can also cooperate well with existing NAS approaches.

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

## A APPENDIX

### A.1 CONFIGURATION DETAILS

All our search experiments are done under the same 11GB VRAM constraint on one RTX 2080Ti GPU.

For the configuration of experiment on CIFAR10, following DARTS, we search for the architecture in a cell manner, where each cell has four intermediate nodes and 16 initial channels during the search. We use half the images (25K) from the training set for kernel weight updating and the other half for architecture parameter updating. When evaluating, we stack 18 normal and two reduction cells to form the entire network.

For the configuration of experiment on ImageNet, when searching, we have two configurations. In the first configuration, following PC-DARTS, we only pick $10\%$ and $2.5\%$ of images from the training set for the kernel weights and architecture parameters' optimizations respectively; in the second configuration, thanks to FTSO's huge computational cost reduction, it is feasible for us to search with the whole of ImageNet. For evaluation, we stack 12 normal cells and two reduction cells to form the entire network and train it from scratch for 250 epochs. The detailed configurations are shown in Table 4.

Table 4: FTSO's searching configurations on CIFAR10 and ImageNet

| Hyper-parameter | Search Config. | | Evaluation Config. | |
|---|---|---|---|---|
| | **CIFAR10** | **ImageNet** | **CIFAR10** | **ImageNet** |
| Cells (Normal + Reduction) | $6 + 2$ | $6 + 2$ | $18 + 2$ | $12 + 2$ |
| Channel proportion | 25% | 25% | - | - |
| Initial channel number | 16 | 16 | 36 | 48 |
| Kernel optimizer | SGDM | SGDM | SGDM | SGDM |
| Kernel learning rate | Cos. $(0.025, 0)$ | 0.5 | Cos. $(0.025, 0)$ | Linear $(0.5, 0)$ |
| Architecture optimizer | Adam | Adam | - | - |
| Architecture learning rate | $6 \times 10^{-4}$ | $6 \times 10^{-3}$ | - | - |

[*] Some items are '-' because, when evaluating, no architecture parameters need to be updated.

