# OpenReview forum: "FTSO: Effective NAS via First Topology Second Operator"
_ICLR.cc/2021/Conference — Reject_

### Official Review · AnonReviewer3 · 2020-10-26
**The paper needs more experiments and the claims are not well-justified by either empirical evidences or theories.**

**Rating:** 4
**Confidence:** 4

**Review:**

The paper proposes a method for Neural Architecture Search (NAS) with two stages of search. In the first  stage, the topology of the cell is searched with only one operator (skip connection) using graph pruning through gradient descents. In the later stage, there are two ways to search the operators. In the first approach, the found topology is equipped with some operators (e.g., 3x3 convolution,  skip connection, and 3x3 dilated convolution) and then the architecture parameters are optimized. Another approach is to replace all operators with one single operator e.g. convolution. The experiments show that the searching time is reduced significantly compared to DARTS and the results on CIFAR-10 and ImageNet are very competitive.


Strengths:

- The search time is reduced significantly from normal DARTS and PC-DARTS.
- The results are competitive to the prior differentiable NAS methods with less searching time.

Weaknesses:

- It is not very clear how exactly the topology search algorithm ensures that each node is connected?
- The choice of skip connection as the operator for topology search is not well-justified. Max-pooling has no learnable parameters too. Is there any particular reason why skip connection?
- The experiments are lacking because there is no evaluation on current benchmarks such as NAS-101, NAs1Shot1, or NAS-201 that can explicitly show how the search algorithm performs. In current experimental results, it is very difficult to judge if the proposed approach has benefits for searching 'good' architectures.
- The comparison with other differentiable NAS algorithms (e.g., DARTS and PC-DARTS) can be delivered in the form of a table. The information of this comparison is all scattered in the manuscript.
- In Section 3 "In FTSO, this problem is almost nonexistent because, the skip connection has no kernel
weights to tune", this is not very clear how the problem exists in the DARTS formulation and how the formulation in the proposed approach can mitigate this issue. There is no discussion about the bi-level program in DARTS.
- It is very difficult to follow the overall approach, it would be better to have an algorithm/pseudocode.
- ", we take the second strategy as the default configuration because, it is much more efficient, and the large amount of kernel weights, contained in the first strategy, may lead to over-fitting,
and finally leads to worse performance than the second strategy. ". How do exactly large amount of kernel weights lead to overfit in NAS? Is there any proof/citation about it? The second strategy has no clear evidence to always give better results compared to the first strategy.
- There is no empirical results how the algorithm works by varying the number of nodes in a cell. This will show that the limit of the proposed approach and also its capacity to handle various number of nodes.

- The manuscript is poorly written,  at least these sentences/paragraphs must be clarified and revised:
** In Section 3,  "it might be possible to cluster the architectures according to their connection topologies"
** In Subsection 4.3, "When we do not search for the operators, after the topology search, we assign all the remaining
edges a fixed operator" and the rest of the paragraph is difficult to follow.
** In Section 3, "Secondly. Because our supernet contains very few parameters, it can hardly over-fit the dataset. Thus, the found architecture may generalize better on new datasets.". This sentence is a claim without any evidence.
** Table 3 is not clear, some codes are not described in its caption.
** In Section 3, please fix "Firstly. It allows the algorithm to converge within extremely few iterations... and so on.."

---

> ### Author Response · Authors · 2020-11-24
> **We have revised the paper following your comments. This is the first part of our reply.**
>
> Thanks for your careful and valuable comments. We will explain your concerns point by point.
>
> Q1: It is not very clear how exactly the topology search algorithm ensures that each node is connected?
> A1: In our topology search, we force every intermediate node to connect to all its previous nodes. Thus, when searching, all the nodes are connected. At the same time, for every intermediate node, we tune its connection weights to all its previous nodes via back-propagation. After the topology search terminates, for every intermediate node, we retain its connections to two previous nodes. This strategy guarantees that we can always find a path from the input node to any node in the graph. In fact, since the only difference between our topology search and the vanilla DARTS[1] is the number of candidate operators, the pruned architecture's connectivity can be guaranteed.
>
> Q2: The choice of skip connection as the operator for topology search is not well-justified. Max-pooling has no learnable parameters too. Is there any particular reason why skip connection?
> A2: As we have shown in Table 3, the max-pooling operator can also bring satisfying results for the topology search. Here, we prefer skip connections because of two reasons. The first reason is that the skip connection operator not only requires 0 parameter, but also produces the minimum computational cost. The second reason is that max pooling greatly reduces the size of the image. Thus, we may lose too much useful information contained in the input image if the network is deep.
>
> Q3: The experiments are lacking because there is no evaluation on current benchmarks such as NAS-101, NAs1Shot1, or NAS-201 that can explicitly show how the search algorithm performs. In current experimental results, it is very difficult to judge if the proposed approach has benefits for searching 'good' architectures.
> A3: We implement our topology search strategy on NATS-Bench (the updated version of NASBench-201) based on their official DARTS code and we have released our code in the attachment. As we mentioned in Section 4.3, according to NATS-Bench’s official documents, the architecture found by DARTS is ‘|skip_connect\~0|+|skip_connect\~0|skip_connect\~1|+|skip_connect\~0|skip_connect\~1|skip_connect\~2|’, in which all the operators are skip connections. This might be a surprising fact. However, it is proven to be true by our reimplementation. This simple architecture provides as low as 54.30% testing accuracy on CIFAR10. For comparison, the architecture found by our topology search is ‘|nor_conv_3x3\~0|+|nor_conv_3x3\~0|nor_conv_3x3\~1|+|nor_conv_3x3\~0|nor_conv_3x3\~1|nor_conv_3x3\~2|’, which can achieve 93.76% testing accuracy on CIFAR10.
>
> Q4: The comparison with other differentiable NAS algorithms (e.g., DARTS and PC-DARTS[2]) can be delivered in the form of a table. The information of this comparison is all scattered in the manuscript.
> A4: Thank you for your advice. We have revised our paper according to your suggestions.
>
> [1] Liu, H., Simonyan, K., & Yang, Y. (2018). Darts: Differentiable architecture search. arXiv preprint arXiv:1806.09055.
> [2] Xu, Y., Xie, L., Zhang, X., Chen, X., Qi, G. J., Tian, Q., & Xiong, H. (2019). Pc-darts: Partial channel connections for memory-efficient differentiable architecture search. arXiv preprint arXiv:1907.05737.

---

> ### Author Response · Authors · 2020-11-24
> **We have revised the paper following your comments. This is the second part of our reply.**
>
> Q5: In Section 3 "In FTSO, this problem is almost nonexistent because, the skip connection has no kernel weights to tune", this is not very clear how the problem exists in the DARTS formulation and how the formulation in the proposed approach can mitigate this issue. There is no discussion about the bi-level program in DARTS.
> A5: In DARTS, we use gradients to update architectures. However, to reduce the computational cost, DARTS does not evaluate the architectures after they get fully converged. Instead, DARTS gives each architecture only one iteration to tune its kernel weights. At the same time, the powerful operators, e.g., convolutions, require large number of gradient updates to get well optimized, and cannot produce meaningful outputs until their kernel weights are well tuned. Within one iteration, only those operators with zero or very few parameters can converge and achieve a relatively satisfying performance. For example, the output of a skip connection operator always equals to its input. Thus, the information contained in the input image can be well retained. Since the simpler operators can retain more meaningful information and perform better than the complex operators when the optimization is inadequate, they obtain greater architecture weights from the gradient updates and contribute more to the network’s output. The result is that these simpler operators gain greater gradients and then get optimized better. Thus, in the end, the simpler operators will have the highest architecture weights and the found architectures tend to only contain the simplest operators. However, we all know that an architecture only containing skip connections cannot achieve high testing accuracy. This is the reason why DARTS cannot perform well. In contrast, our topology search utilizes only skip connections, an operator with no kernel weight. Thus, in FSTO, the super-net can converge within zero iteration. In this case, we can measure every architecture’s performance precisely and reduce the bi-level optimization problem to single level, which has been widely studied and is theoretically guaranteed to be convergeable given an appropriate learning rate. This is the reason why we say that this problem is almost nonexistent in FTSO.
>
> Q6: It is very difficult to follow the overall approach, it would be better to have an algorithm/pseudocode.
> A6: Thank you for your advice. We have revised our paper according to your suggestions. Please check Algorithm 1 and 2 for more details.
>
> Q7: ", we take the second strategy as the default configuration because, it is much more efficient, and the large amount of kernel weights, contained in the first strategy, may lead to over-fitting, and finally leads to worse performance than the second strategy. ". How does exactly large amount of kernel weights lead to overfit in NAS? Is there any proof/citation about it? The second strategy has no clear evidence to always give better results compared to the first strategy.
> A7: Large amount of kernel weights lead to overfitting in DARTS because the goal of DARTS is to find a sub-graph from a huge super-net. Suppose the sub-graph can generalize perfectly on the testing set. Since the sub-graph is only part of the super-net, and the super-net contains many times more parameters than the sub-graph, the super-net must over-fit the dataset and cannot generalize well on the testing set. This is the reason why we say that large amount of kernel weights leads to overfitting in DARTS. PC-DARTS has mentioned this issue and reduces the size of the super-net via only computing partial input channels. For the third question, we did not say that the second strategy can always give better results than the first strategy. In fact, our opinion is that the second strategy is simpler, and it can stably provide architectures with comparable performance because, by replacing the simpler operators with complex operators in the sub-graph, the found architecture’s model capacity gets promoted. For comparison, the first strategy produces a huge super-net containing relatively large number of kernel weights, thus suffers from similar overfitting problems in DARTS.

---

> ### Author Response · Authors · 2020-11-24
> **We have revised the paper following your comments. This is the third part of our reply.**
>
> Q8: There is no empirical results how the algorithm works by varying the number of nodes in a cell. This will show that the limit of the proposed approach and also its capacity to handle various number of nodes.
> A8: As we have shown in A3, FTSO can greatly surpass the baseline on NATS-Bench as well. This search space contains 5 nodes, which is different to our standard configuration with 7 nodes. This result clearly shows FTSO’s generalizability on different search spaces and nodes. Theoretically, while the number of nodes increases, the size of the super-net dramatically increases. This leads to much more severe overfitting and computational cost in DARTS. However, in FTSO, we ease the overfitting via forming a super-net with only skip connections and reduce the computational cost from $O(n^3)$ to $O(n^2)$ as mentioned in Section 3. Thus, FTSO’s advantages to DARTS can be even larger when dealing with more nodes. Although we can easily extend FTSO to more nodes, it is impractical to implement DARTS with more nodes due to DARTS’s unbearable computational cost. This is the reason why we cannot provide an empirical comparison to DARTS with more nodes.
>
> Q9: The manuscript is poorly written, at least these sentences/paragraphs must be clarified and revised. ** In Section 3, "it might be possible to cluster the architectures according to their connection topologies" ** In Subsection 4.3, "When we do not search for the operators, after the topology search, we assign all the remaining edges a fixed operator" and the rest of the paragraph is difficult to follow. ** In Section 3, "Secondly. Because our supernet contains very few parameters, it can hardly over-fit the dataset. Thus, the found architecture may generalize better on new datasets.". This sentence is a claim without any evidence. ** Table 3 is not clear, some codes are not described in its caption. ** In Section 3, please fix "Firstly. It allows the algorithm to converge within extremely few iterations... and so on.."
> A9: Thank you for your advice. We have revised our paper according to your suggestions.

---

### Official Review · AnonReviewer1 · 2020-10-29
**An interesting paper with very promising result**

**Rating:** 5
**Confidence:** 3

**Review:**

This work researches the issue of neural architecture search (NAS), which is of significance for practical applications of deep neural networks and has become an active research topic in the past several years. Many methods on NAS have been developed recently. The computational efficiency of search has been one of the obstacles for this line of research.

Strengths of this work:

The key idea of this work is to decouple the search of network topology and the search of operators. By doing so, the computational efficiency can be substantially increased (say, reducing the search time from several GPU-days to less than one second), while well maintaining the classification performance or even slightly improving it. Experimental study on CIFAR10 and ImageNet demonstrates the advantage of the proposed method, especially the improvement on computational cost. The paper is overall well written.

Weaknesses:

1. This work can do better on theoretical analysis. Currently, it mainly describes how the decoupled search (FTSO) is implemented (in Section 3). Considering the significance of the improvement on computational efficiency, it will be more valuable to discuss whether such a kind of decoupled search can be generally applied, or it can only work for certain kinds of network architecture, data, or tasks. This will provide more insights on this interesting method and theoretical contribution.

2. This paper has described several interesting observations. It will be better if they are further explained and discussed.

2-1. When conducting topology search, this paper only uses a simple operator (say, skip connection). It is indicated that this supernet can hardly overfit the dataset, which is understandable. Meanwhile, with such a simple operator, will the supernet underfit the data? This needs to be clarified.

2-2. Similarly, in Section 4.1, it is stated that FTSO contains very few parameters and therefore can even achieve comparable performance to PC-DARTS. Why could very few parameters lead to this comparable performance? Please clarify it.

2-3. At the end of Section 4.2, it is found that the best architecture obtained by the search is the shallowest and widest one. The current explanation (provided at the end of this Section) is too brief and vague for such an interesting and surprising finding. This need to be further clarified.

2-4. The part at the top of page 7 discusses the performance with respect to iteration and epoch. It is not very clear. It is stated that "although one iteration cannot surpass one epoch, it is better than a few iterations." However, considering that one epoch consists of multiple iterations, this statement seems to contradict to itself and is a bit hard to follow. Please clarify.

3. The second paragraph on page 4 compares the computational complexity of DARTS and the proposed method. Since this is the key part of this work, more details shall be provided on how the complexity is worked out.

4. Minor issues:

4-1. The first sentence of the second paragraph on page 2 needs to be revised.
4-2. Although the investigation of correlation among different architectures is mentioned as the key motivation for this work, how is the correlation of architectures considered in this work is not presented. This shall be improved;
4-3. In Section 4.2, the resolution of images in ImageNet is reduced to 28x28. Does this imply that the proposed method is not effective enough to deal with images of normal size?
4-4. Figures 1 and 2 shall be enlarged.

--- Thank the authors for the detailed response. After reading the response and the comments of peer reviewers, the rating is altered as follows.

---

> ### Author Response · Authors · 2020-11-24
> **We have revised the paper following your comments. This is the first part of our reply.**
>
> Thanks for your careful and valuable comments. We will explain your concerns point by point.
>
> Q1: This work can do better on theoretical analysis. Currently, it mainly describes how the decoupled search (FTSO) is implemented (in Section 3). Considering the significance of the improvement on computational efficiency, it will be more valuable to discuss whether such a kind of decoupled search can be generally applied, or it can only work for certain kinds of network architecture, data, or tasks. This will provide more insights on this interesting method and theoretical contribution.
> A1: As we have mentioned in Section 5 and the response to Reviewer 3’s Q3, FTSO is not only superior on DARTS’s default search space, but also greatly dominating DARTS[1] on NATS-Bench dataset. Although currently we have only tested FTSO on image classification tasks, since nearly all the vision algorithms are based on image classification models, we have reasons to believe that FTSO can perform well on most vision tasks. At the same time, we have also discussed how to transfer FTSO to other machine learning areas in Section 5. Although, due to the limitation of time and resources, we did not test our method on other areas. However, we plan to make up experiments on language recognition in the future.
>
> Q2-1: When conducting topology search, this paper only uses a simple operator (say, skip connection). It is indicated that this supernet can hardly overfit the dataset, which is understandable. Meanwhile, with such a simple operator, will the supernet underfit the data? This needs to be clarified.
> A2-1: It is true that the super-net containing only skip connections underfits the data. However, this is indeed our desire because, if this simple super-net already perfectly fits the data, then after replacing the skip connections with more complex operators, the finally exported architecture will absolutely overfits the data. What is more, when extracting a discrete sub-graph from the super-net, many operators with architecture weights lager than 0 are dropped. This is equivalent to set some architecture parameters to 0. For a super-net overfitting the data, it tends to find a sharp local minimum. While the super-net underfitting the data prefers a flat minimum. Thus, suppose we have found the global optimums of the both the overfitting super-net and the underfitting super-net, the introduced disturbance will impact more on the overfitting super-net. This is the reason why we say an underfitting super-net can produce better architectures.
>
> Q2-2: Similarly, in Section 4.1, it is stated that FTSO contains very few parameters and therefore can even achieve comparable performance to PC-DARTS[2]. Why could very few parameters lead to this comparable performance? Please clarify it.
> A2-2: This is because when we say FTSO outperforms PC-DARTS, we mean that the model we find with FTSO can achieve higher testing accuracy than PC-DARTS. As we have mentioned in Q2-1, the parameter number of PC-DARTS’s super-net is much larger than what is needed. Thus, few parameters of FTSO lead to higher performance.
>
> Q2-3: At the end of Section 4.2, it is found that the best architecture obtained by the search is the shallowest and widest one. The current explanation (provided at the end of this Section) is too brief and vague for such an interesting and surprising finding. This need to be further clarified.
> A2-3: We have discussed more about this issue in the revised paper. We think the main reason is that the whole model is stacked with many cells. If the depth of each cell is too high, it leads to a very deep neural network. At that time, because all the operators in our found architecture are convolutions, we cannot use skip connections to facilitate gradients’ propagation in ResNet’s manner. In this way, both the vanishing and explosion of gradients may prevent the deeper models from higher performance.
>
> [1] Liu, H., Simonyan, K., & Yang, Y. (2018). Darts: Differentiable architecture search. arXiv preprint arXiv:1806.09055.
> [2] Xu, Y., Xie, L., Zhang, X., Chen, X., Qi, G. J., Tian, Q., & Xiong, H. (2019). Pc-darts: Partial channel connections for memory-efficient differentiable architecture search. arXiv preprint arXiv:1907.05737.

---

> ### Author Response · Authors · 2020-11-24
> **We have revised the paper following your comments. This is the second part of our reply.**
>
> Q2-4: The part at the top of page 7 discusses the performance with respect to iteration and epoch. It is not very clear. It is stated that "although one iteration cannot surpass one epoch, it is better than a few iterations." However, considering that one epoch consists of multiple iterations, this statement seems to contradict to itself and is a bit hard to follow. Please clarify.
> A2-4: This might be a little counter-intuitive. However, it coincides with our experiment results. The reason is that when we only search for one iteration, it is obvious that the overfitting cannot happen. At the same time, because the super-net contains only skip connections, it does not need many iterations to converge. This is the reason that one iteration can perform well. When we search for a few iterations, the number of different images seen by the model is not big enough. However, since the super-net is very small, such number of gradient updates may have been enough for the super-net to get overfitted on the training set. This is the reason why a few iterations perform may worse than one or two iterations. After we have searched for one whole epoch, the super-net has seen enormous different images, this helps it to generalize better on the testing set. This is the reason why one epoch performs the best.
>
> Q3: The second paragraph on page 4 compares the computational complexity of DARTS and the proposed method. Since this is the key part of this work, more details shall be provided on how the complexity is worked out.
> A3: Thank you for your advice. We have presented more detailed derivations in the revised paper’s Section 3.
>
> Q4-1: The first sentence of the second paragraph on page 2 needs to be revised.
> A4-1: Thank you for your advice. We have revised the paper according to your comments.
>
> Q4-2: Although the investigation of correlation among different architectures is mentioned as the key motivation for this work, how is the correlation of architectures considered in this work is not presented. This shall be improved;
> A4-2: Thank you for your advice. We have explained this in detail in the revised paper. The first correlation is that [3] mentions that randomly varying operators in a good architecture leads to another good architecture. The second correlation is that simple operators can be treated as a special case of complex operators. For example, we can properly set the kernel weights of a convolution to make its output = input. At this time, a convolution performs as a skip connection. Thus, if a network with only skip connections performs well, it means that when substituting all the skip connections with convolutions, at least we can find a set of weights to ensure the new network performs well. In other words, suppose we define all the architectures having the same operators yet different topologies with an architecture A as A’s neighbors. If A is an architecture with top accuracy among all its neighbors and B has the same topology as A, then B will also be an architecture with top accuracy among all its neighbors. The way we utilize this correlation is to first form a super-net only containing skip connections, then search for architectures on the super-net. After we found the best architecture on the shrunk super-net, we replace all its operators with convolutions to generate the final architecture.
>
> Q4-3: In Section 4.2, the resolution of images in ImageNet is reduced to 28x28. Does this imply that the proposed method is not effective enough to deal with images of normal size?
> A4-3: This configuration is inherited from the baseline, namely PC-DARTS. In fact, while dealing with larger images, FTSO has more superiority because it avoids complex operations on the input images. Suppose we enlarge the image resolution from 28x28 to 280x280, then, the computational cost of every operator in our candidate operator set will be 100 times greater. At this time, PC-DARTS’s time consumption will be about 380 GPU-days, while FTSO only requires about 1 GPU-minute.
>
> Q4-4: Figures 1 and 2 shall be enlarged.
> A4-4: Thank you for your advice. We have revised the paper according to your comments.
>
> [3] Shu, Y., Wang, W., & Cai, S. (2019, September). Understanding Architectures Learnt by Cell-based Neural Architecture Search. In International Conference on Learning Representations.

---

### Official Review · AnonReviewer4 · 2020-10-31
**inadequate presentation quality**

**Rating:** 3
**Confidence:** 5

**Review:**

Title: FTSO: EFFECTIVE NAS VIA FIRST TOPOLOGY SECOND OPERATOR

Summary of the paper:

The goal is to fit the hyper-parameters of the neural network namely topology (i.e. network architecture) and operator (e.g. skip connection or convolution) rather than just fit the parameters (i.e. weights). The approach is similar to DARTS which relaxes the architecture choice to obtain a continuous optimisation.

The main difference from DARTS is that rather than jointly selecting operator and topology, first the topology and then the operator is selected. It's roughly a form of coordinate descent.

The paper is empirical in nature and claims good results.

Pros:

The idea of optimising topology then operator appears to work well in practice.

Cons:

The presentation is far from top tier conference standard.

Examples include, this sentence from the introduction (yes, it's one sentence):

"We first mathematically prove that, by greatly shrinking the graph of the search space, reducing the operators’ complexity in magnitude, lowering the required searching period from 50 epochs to one iteration and significantly easing the Matthew effect, namely that the complex operators may never get the chance to be well tuned, thus the found architecture only contains very simple operators, and performs poorly on the testing set, FTSO reduces the required parameters by a factor of 0.53×108, decreases the FLOPs per iteration by a factor of 2×105 and significantly promotes the accuracy compared to the baseline, PC-DARTS."

(Anyway, the "mathematical proof" aluded to is an informal argument for an example, essentially saying that searching one coordinate then the other is cheaper than searching both jointly.)

Also figures 2 and 3 are unreadable on printout.

Also, there is no code to check the results. This is a significant factor considering the empirical nature.

Recommendation:

The paper is surely interesting to some, but needs a lot more work.

---

> ### Author Response · Authors · 2020-11-24
> **We have polished the draft and updated the revised paper.**
>
> Thanks for your careful and valuable comments. We will explain your concerns point by point.
>
> Q1: The presentation is far from top tier conference standard. Examples include this sentence from the introduction.
> A1: Thank you for your advice. We have polished the draft and updated the revised paper.
>
> Q2: Anyway, the "mathematical proof" aluded to is an informal argument for an example, essentially saying that searching one coordinate then the other is cheaper than searching both jointly.
> A2: We illustrate the derivation more detailedly in the revised paper's Section 3. As a general conclusion, DARTS[1] requires $\frac{1}{2}pn(n-3)H_{out}W_{out}C_{out}(k^2C_{in}+1)$ FLOPs and $\frac{1}{2}n(n-3)p(k^2C_{in}+1)C_{out}$ parameters, while the total number of FLOPs and parameters of FTSO are $\frac{1}{2}n(n-3)H_{in}W_{in}C_{in}$ and $\frac{1}{2}n(n-3)$ respectively. Here $k$ is the kernel size, $n$ is the number of nodes, $p$ is the number of candidate operators, $C_{in}$ is the input tensor's channel number, $H_{out}$, $W_{out}$, and $C_{out}$ are the output tensor's height, width and channel number respectively. To show this result more intuitively, we assign the parameters in the formulas a set of values commonly used in NAS papers. In this concrete example, FTSO requires only $\frac{1}{p(k^2C_{in}+1)C_{out}}=1.9\times 10^{-8}$ times the parameters and $\frac{1}{p(k^2C_{in}+1)}=9.8\times 10^{-6}$ times the forward-propagation FLOPs per iteration compared to DARTS.
>
> Q3: Figures 2 and 3 are unreadable on printout.
> A3: Thank you for your advice. We have enlarged Figure 2 and 3 in the revised paper.
>
>
> Q4: There is no code to check the results. This is a significant factor considering the empirical nature.
> A4: Thank you for your advice. We have uploaded the code as the supplementary material
>
> [1] Liu, H., Simonyan, K., & Yang, Y. (2018). Darts: Differentiable architecture search. arXiv preprint arXiv:1806.09055.

---

### Decision · Program_Chairs · 2021-01-07
**Final Decision**

**Decision:**

Reject

**Comment:**

Three reviewers have reviewed this manuscript, and they had severe reservations regarding the presentation quality and the lack of sufficient theoretical support behind empirical observations. Even after rebuttal, the reviewers maintained that the above issues are not fully resolved. Unfortunately, this paper cannot be accepted in its current form.